# Acoustic spin-Chern insulator induced by synthetic spin–orbit coupling with spin conservation breaking

Weiyin Deng [1,5], Xueqin Huang [1,5], Jiuyang Lu [1], Valerio Peri [2], Feng Li [1✉], Sebastian D. Huber [2] & Zhengyou Liu [3,4✉]

Topologically protected surface modes of classical waves hold the promise to enable a variety of applications ranging from robust transport of energy to reliable information processing networks. However, both the route of implementing an analogue of the quantum Hall effect as well as the quantum spin Hall effect are obstructed for acoustics by the requirement of a magnetic field, or the presence of fermionic quantum statistics, respectively. Here, we construct a two-dimensional topological acoustic crystal induced by the synthetic spin-orbit coupling, a crucial ingredient of topological insulators, with spin non-conservation. Our setup allows us to free ourselves of symmetry constraints as we rely on the concept of a non-vanishing "spin" Chern number. We experimentally characterize the emerging boundary states which we show to be gapless and helical. More importantly, we observe the spin flipping transport in an H-shaped device, demonstrating evidently the spin non-conservation of the boundary states.

[1] School of Physics and Optoelectronics, South China University of Technology, Guangzhou, Guangdong 510640, China. [2] Institute for Theoretical Physics, ETH Zurich, Zürich 8093, Switzerland. [3] Key Laboratory of Artificial Micro- and Nanostructures of Ministry of Education and School of Physics and Technology, Wuhan University, Wuhan 430072, China. [4] Institute for Advanced Studies, Wuhan University, Wuhan 430072, China. [5]These authors contributed equally: Weiyin Deng, Xueqin Huang. ✉email: phlifeng@scut.edu.cn; zyliu@whu.edu.cn

The discovery of topological insulators (TIs), featuring a bulk gap and gapless boundary states, opened new avenues for condensed-matter physics[1,2]. In two spatial dimensions, TIs come in two different classes, either described by a Z or $Z_2$ topological index. The first[3–5] breaks time-reversal symmetry (TRS), and are commonly called Chern insulators (CIs). They host an (anomalous) integer quantum Hall effect and their surfaces are characterized by chiral, i.e., unidirectional, surface states. The latter $Z_2$ insulators[6–8], such as the quantum spin Hall effect, are characterized by a pair of gapless helical boundary states. In the presence of spin conservation, the $Z_2$ insulators can equivalently be described by spin-Chern numbers, where the spin sectors might carry an opposite but nonzero Chern number[9]. In fact, spin-Chern numbers are well defined even in the absence of spin conservation or for TRS-broken systems[9–11]. The spin-Chern numbers have been employed to identify TRS-broken spin-1/2 electronic TIs and pseudospin TIs, giving rise to the concept of spin-Chern insulators (SCIs)[10–17]. The SCIs feature helical boundary states, but whether gapless or not, depends on the system symmetry and microstructure of the sample boundary[17].

Recently, intense efforts have been devoted to realizing classical analogs of TIs for electromagnetic, mechanical, and acoustic waves[18–21]. Photonic CIs have been realized in magneto-optic systems[20–23], mechanical CIs have been proposed in gyroscopic metamaterials[24,25], and acoustic CIs have been proposed in systems with circulating fluid[26–29] and experimentally implemented recently[30]. Hafezi et al. achieved a photonic SCI and observed the helical boundary states in a silicon photonic crystal. The pseudospin–orbit coupling was induced by the differential optical paths based on ring resonators[31,32]. Mechanical SCIs were realized in bilayer structures, which rely on opposite interlayer and intralayer couplings[33,34].

Helical edge states have been observed in acoustic systems[35–44]. Although these systems all host pseudospin, since the spin–orbit coupling, the essential ingredient for a SCI, is not available, they are not the acoustic SCIs. This can be reflected by fact that the helical edge states in the systems exist on the domain walls or interfaces, rather than on the boundaries or surfaces, as in SCIs. Returning to the photonic or mechanical SCIs aforementioned[31–34], it can be noted that the pseudospin in the SCIs is conserved, which means that these SCIs can actually be viewed as two independent copies of CIs and the topological properties can be described by Chern numbers. When the pseudospin conservation is broken, the description in terms of Chern numbers is no longer valid. One then needs to rely on the spin-Chern number to characterize the SCIs, in which the spin can vary or flip during transport. However, this general case has not yet been explored in all classical scenarios. A natural question arises: can we achieve acoustic SCIs (with spin–orbit coupling) without spin conservation?

In this work, to answer this question, we realize an acoustic spin-Chern insulator (ASCI) in a bilayer phononic crystal (PC). By introducing a layer pseudospin degree of freedom and the proper interlayer coupling, a synthetic spin–orbit interaction is successfully induced, which, particularly, also breaks the pseudospin conservation. We will first introduce the tight-binding model built on a bilayer Lieb lattice, which hosts all the physics of the ASCI. Then we will map this discrete model to a practical PC and demonstrate the topological properties of the ASCI, including the robust, transport of the helical or spin-momentum locking boundary states in the ASCI. In particular, we will present the observation of the spin-flipping transport in a H-shaped device, evidencing the spin nonconservation of the boundary states.

## Results

**Tight-binding model for SCI.** To illustrate how to realize an ASCI, we construct a tight-binding model on a bilayer Lieb lattice

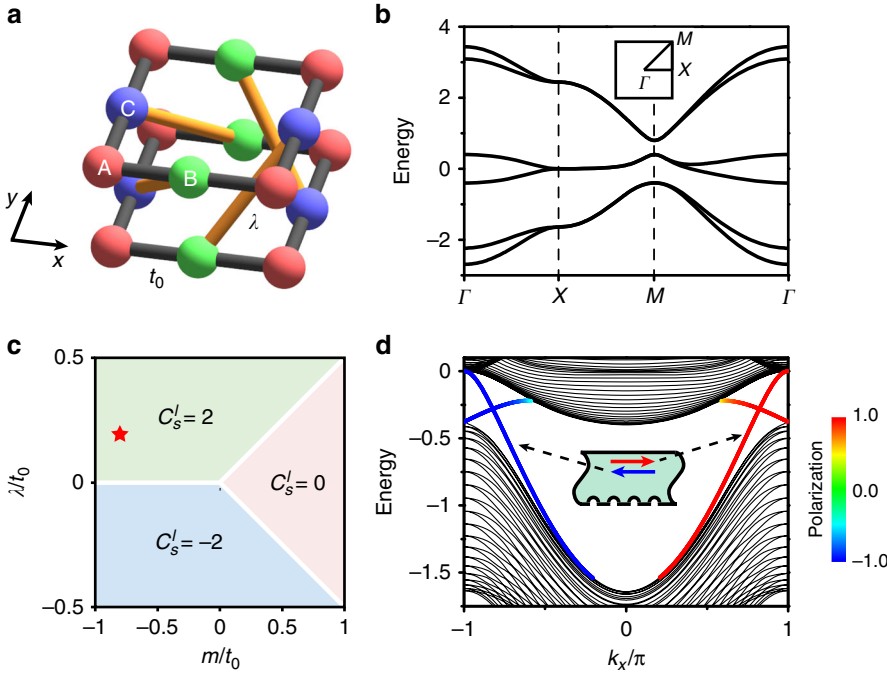

**Fig. 1 Spin-Chern insulator and helical boundary states for a bilayer Lieb lattice model. a** Schematic of the lattice structure. The red, green, and blue spheres of each layer denote A, B, and C lattices. **b** The bulk band structure along high symmetry lines. The interlayer coupling gives rise to two band gaps. Inset: the first Brillouin zone. **c** Phase diagram determined by the spin-Chern number of the lower two bands $C_s^l$ in the $\lambda/t_0$ and $m/t_0$ plane. The white lines represent lower bandgap closure. The red star denotes the phase with the specific parameters used in **b** and **d**. **d** The boundary-state dispersion of a ribbon with $C_s^l = 2$ in the lower gap. A pair of boundary states at the same edge (inset) have opposite layer pseudospin polarizations (red and blue colors). The parameters are chosen as $t_0 = -1$, $\lambda = -0.2$, and $m = 0.8$.

with a unit cell containing three sites in each layer, denoted A (red sphere), B (green sphere), and C (blue sphere) in Fig. 1a. The Hamiltonian is

$$H = t_0 \sum_{\langle ij \rangle, \alpha} c_{i\alpha}^\dagger c_{j\alpha} + m \sum_{i \in A, \alpha} c_{i\alpha}^\dagger c_{i\alpha} + \lambda \sum_{\langle\langle ij \rangle\rangle, \alpha \neq \beta} v_{ij,\alpha} c_{i\alpha}^\dagger c_{j\beta}, \qquad (1)$$

where $c_{i\alpha}^\dagger$ is the creation operator of layer pseudospin $\alpha$ on site $i$. The first term describes the nearest-neighbor intralayer hopping with strength $t_0$. The second term denotes the on-site energy $m$ on site A. The last term represents the chiral interlayer coupling with strength $\lambda$, where $v_{ij,\alpha} = [\varepsilon_\alpha (\hat{e}_{kj} \times \hat{e}_{ik})_z + 1]/2$ with $\varepsilon_{\uparrow\downarrow} = \pm 1$, where $i$ and $j$ are two next-nearest-neighbor sites with $i \neq j$, $k$ is their unique common nearest-neighbor site, and the unit vector $\hat{e}_{kj}$ points from $j$ to $k$ (see Supplementary Note 1 for details). The band dispersion of the model is presented in Fig. 1b. The interlayer coupling opens two bulk gaps and can induce topological phase transitions in the model.

The topological properties of this system can be captured by a spin-Chern number. One can introduce a pseudospin $\tau_\alpha = \sigma_\alpha \otimes I_3$, where the Pauli matrices $\sigma_\alpha$ act on the layer degree of freedom. While none of the components of $\tau$ is conserved, one can use the projection of, say $\sigma_y$ into pairs of bands below the gap to split them. These split bands lead to well-defined fiber bundles that may carry a nonzero Chern number: the spin-Chern number. These spin-Chern numbers are a tool well tailored to classical systems, as they neither require any symmetry nor the presence of a fermionic time-reversal operator. However, it is important to note that one relies not only on a spectral gap, but also on the spin-projection gap that allows for the splitting of the bands. Moreover, the details of the edge physics have to be inspected independently of the bulk, as there might be a spin-gap closing induced by the surface termination[10,11], see Supplementary Note 2 for details.

For illustrations, we focus on the tological properties of the lower gap. Figure 1c shows the spin-Chern number of the lower two bands $C_s^l$ as a function of $\lambda/t_0$ and $m/t_0$. Three topologically distinct phases exist. At the phase boundaries, indicated by the white lines, the energetic bulk gap closes. In the absence of spin–orbit interaction ($\lambda = 0$, the mass term opens a trivial gap: $C_s^l = 0$). The projected band dispersion for a ribbon with $C_s^l = 2$ is plotted in Fig. 1d. The boundary states of the two lines oriented along the dotted black arrows localize at one boundary of the ribbon (the inset), while the others localize at the other boundary. The complete topological phases, the generalized bulk-boundary correspondence, and associated helical boundary states are shown in Supplementary Note 2–4, respectively.

**The SCI for acoustic waves**. We now consider a PC implementation of the SCI for acoustic waves. As shown in Fig. 2a, the PC sample, fabricated by 3D printing, consists of a bilayer structure with interlayer connections realized by chiral tubes. Each layer of the unit cell contains three nonequivalent cavities connected by intralayer tubes (Fig. 2b). Mapping the PC to a tight-binding model, the cavities can be regarded as lattice sites, while the tubes provide hopping terms. The square unit cell has in-plane length $a = 20$ mm and height $h = 12.5$ mm. The three square cavities composing a layer of the unit cell have the same height $h_c = 5$ mm and different in-plane dimensions: $L_A = 7$ mm and $L_B = L_C = L_0 = 8$ mm. The width and height of the intralayer tubes are $L_t = 3.2$ mm and $h_t = 3$ mm, respectively. The diameter of the interlayer tubes is $d = 3.2$ mm. Since the volume of cavity A is smaller than those of the other cavities, modes localized there are detuned to higher frequencies. This corresponds to a regime

$m/t_0 < 0$ for the tight-binding model of Eq. (1): a region of the phase diagram with $C_s^l = 2$.

In Fig. 2c, we present the measured bulk band dispersion along high symmetry lines. Overlaid to the experimental data, we show the simulated bands. A bulk gap at M opens, thanks to the chiral interlayer couplers. To confirm the topologically nontrivial nature of the gap, we calculate the spin-dependent Berry curvature and spin-Chern number of the lowest two bands in the real PC. As the pressure field is mainly localized at the cavities, we construct the normalized wavefunctions $\varphi(\mathbf{k})$ of the PC by using the pressure field sampled at the center of each cavity. Using these wavefunctions (Supplementary Note 2), we obtain the spin-dependent Berry curvatures $\Omega_\pm^l(\mathbf{k})$, shown in Fig. 2d. The two spin-projection sectors have opposite Berry curvature. By integrating separately $\Omega_\pm^l(\mathbf{k})$ in the whole Brillouin zone, we determine the spin-Chern number of the PC. The result, $C_s^l = 2$, confirms that the PC has a gap with the same nontrivial topology predicted by the tight-binding model. The other phases of the ASCI, corresponding to the cases of $\lambda/t_0 > 0$, are studied in Supplementary Note 5.

**Helical boundary states in an ASCI**. The nonzero $C_s^l$ can induce a pair of helical boundary modes, even in the absence of crystalline symmetries (Supplementary Note 6). The projected band dispersions along the $k_x$ direction are plotted in Fig. 3a and b for the whole-cell and half-cell boundaries, respectively. The color maps represent the experimental dispersions, while the overlaid lines are the result of full-wave simulations. A pair of counter-propagating gapless boundary states (solid white lines) exists in the gap for both boundaries. The spin polarization along the $y$ direction is defined as $\langle \sigma_y \rangle = \langle \psi_k | \sigma_y | \psi_k \rangle$, where $\psi_k$ is the eigenmode of the projected dispersion of the PC ribbon sampled at the center of each cavity. The spin nonconservation of the boundary states results in $|\langle \sigma_y \rangle| < 1$. On the other hand, although the spin is nonconserved, the boundary states still host spin-momentum locking property on each boundary. This is because that the boundary states satisfy $\psi_{-k} = \psi_k^*$ in the presence of TRS, and thus possess opposite spin polarization as $\langle \psi_{-k} | \sigma_y | \psi_{-k} \rangle = -\langle \psi_k | \sigma_y | \psi_k \rangle$. This can be understood as a generalization of the spin-momentum locking for the helical boundary states. These results are demonstrated both by the simulations (the lines) and experiments (circles) in Fig. 3c and d. We note that the $\sigma_y$ for the boundary states with $k_x < 0$ and with $k_x > 0$ are opposite, both satisfying $|\langle \sigma_y \rangle| < 1$.

Although the spin is nonconserved, the boundary waves are still robust against backscattering induced by the defect with TRS, because the spin-momentum locking property leads to destructive interference of the backscattering wave[2]. The acoustic waves are nonreciprocal or non-time-reversal in a flowing fluid[26–30], so a "magnetic defect" for an acoustic system can be created by circulating the fluid component somewhere locally in the structure. For our system, a "magnetic defect" on the boundary may be created by circulating the air inside a certain tube close to the boundary connecting the upper and lower layers. Practically, creating circulating flow brings much complexity in design and in experiment. Therefore, an equivalent "magnetic defect" for acoustic waves is rare, which means that the helical edge states observed in our acoustic system are topologically protected. Figure 4a shows the transport of acoustic boundary wave at 7.44 kHz in a sample possessing a rectangular defect. The boundary waves propagate smoothly around the defect. The experimental (upper panel) result is consistent with the simulation one (lower panel). However, the measured transmission is lower than that by simulation, as shown in Fig. 4b, because the boundary waves

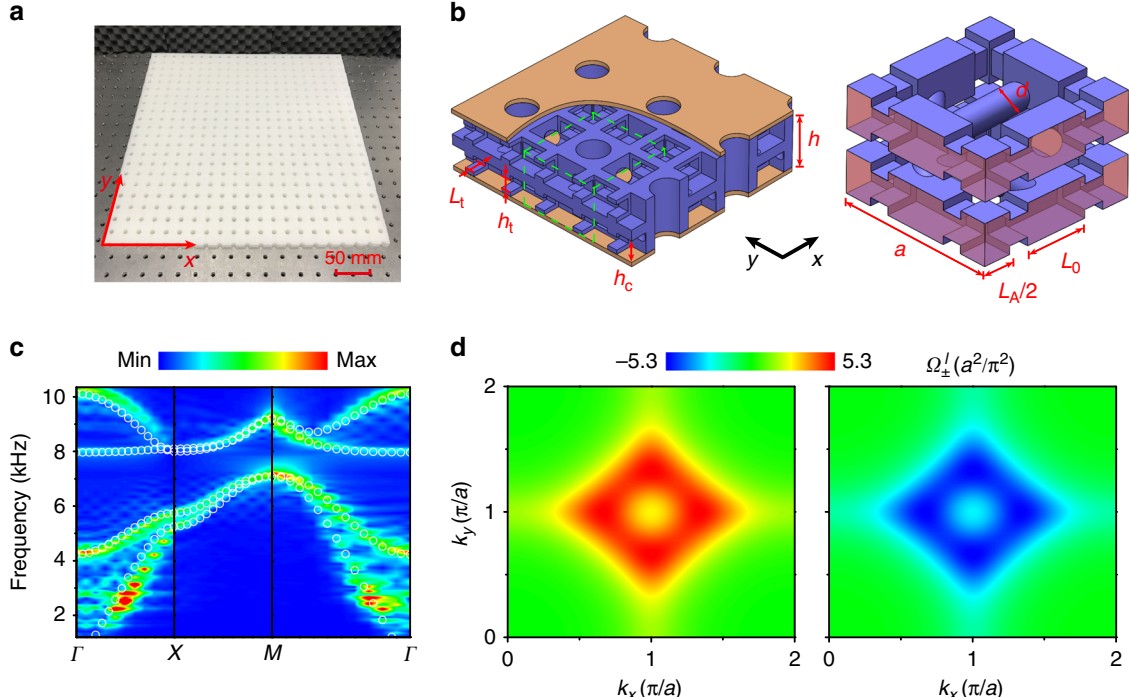

**Fig. 2 Acoustic spin-Chern insulator and bulk properties. a** A photo of the bilayer sample. The cylindrical holes are indeed for cost savings and have no effect on the propagation of acoustic waves in the PC. **b** Left panel: magnified side view of the sample, which contains the whole-cell (up-edge) and half-cell (down-edge) boundaries along the $x$ direction. Air fills the inside of the bilayer structure (blue color) between two rigid plates (brown color). Right panel: the unit cell of the sample, corresponding to the green dotted box in the left panel. The blue (pink) areas represent the rigid (periodic) boundaries. **c** The bulk band structure of the lowest four modes along high symmetry lines. The color maps denote the measured data, and the white circles represent the simulated results. **d** The calculated Berry curvatures of the spin-up (left) and -down (right) projection sectors for the lower two bands.

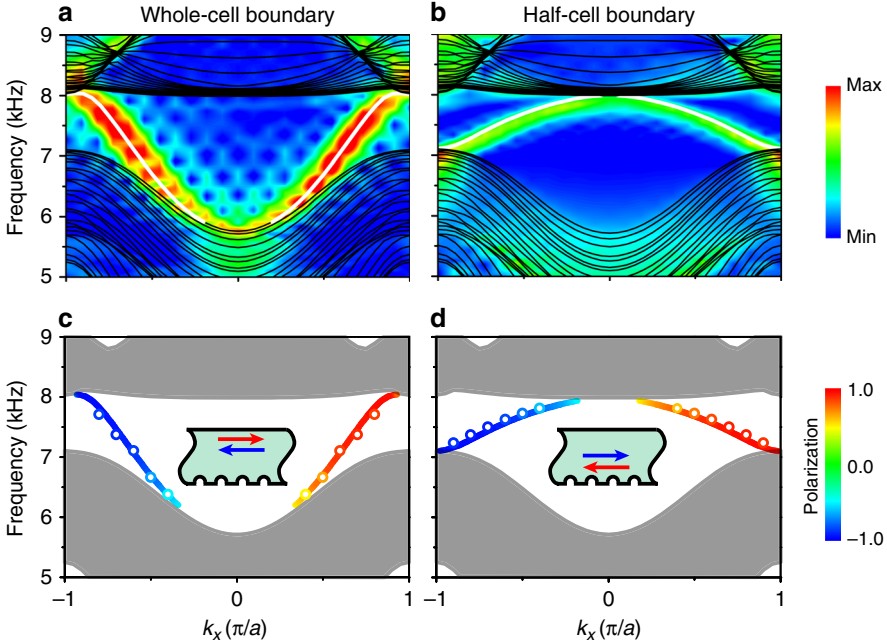

**Fig. 3 Acoustic helical boundary waves. a**, **b** The dispersions of helical boundary waves on the whole-cell and half-cell boundaries, respectively. The color maps denote the measured data, and the white and black lines represent the simulated dispersions of the boundary states and projected bulk states, respectively. **c**, **d** The spin polarizations of the boundary waves for the whole-cell and half-cell boundaries, respectively (lines for simulations and circles for experimental results). Inset: the red (blue) color of arrows denotes spin up (down). A pair of gapless boundary waves with opposite spin polarizations counterpropagate along the boundary.

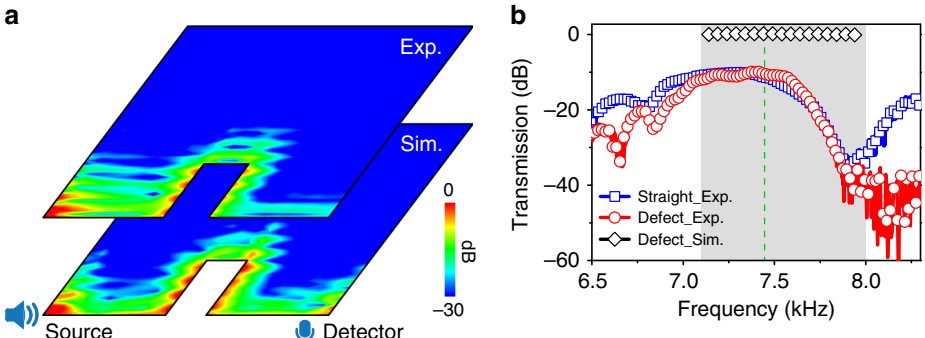

**Fig. 4 Experimental demonstration of backscattering robustness of the acoustic boundary waves for the whole-cell boundary. a** The field distributions 7.44 kHz (dashed line in **b**) from the simulation and experiment for a sample possessing a rectangular defect. The loudspeaker and microphone symbols denote the source and detector positions, respectively. **b** The simulated (diamond) and measured (circle) transmissions for the defect path, compared with measured one, are for a sample having a straight path with the same length. The good agreement of the measured results in the bulk gap (shadowed region) indicates the negligibly weak backscattering of the acoustic boundary waves.

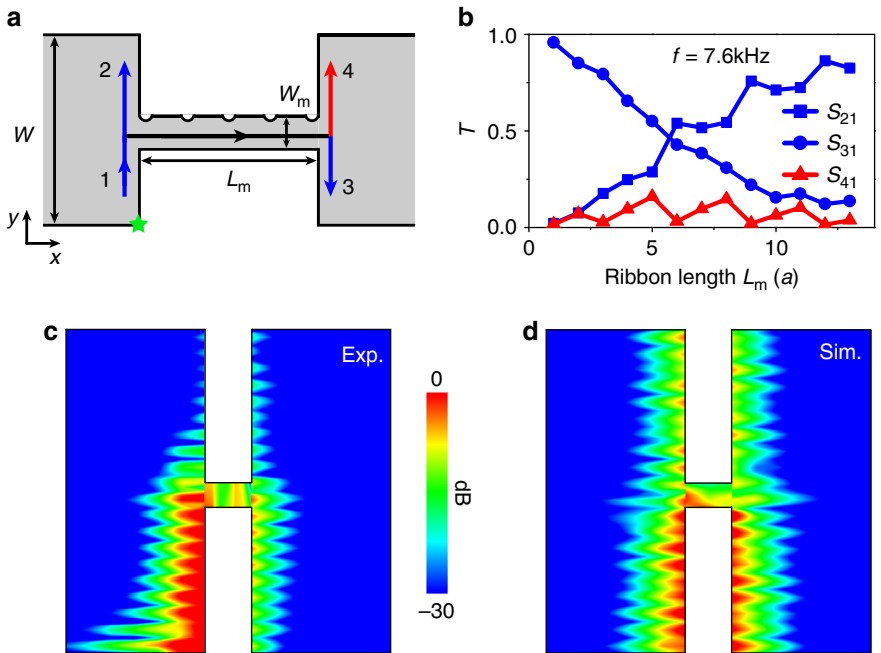

**Fig. 5 The spin flipping. a** The schematic of the H-shaped PC, where the boundary wave with spin-down polarization is excited at channel 1, and propagates to terminals 2 and 3 with the same polarization, but to terminal 4 with spin-up polarization. **b** The calculated transmissions from channel 1 to terminals 2 ($S_{21}$), 3 ($S_{31}$), and 4 ($S_{41}$) in the H-shaped sample. **c**, **d** The experimental and simulated field distributions for $W_m = 1.5a$ and $L_m = 2a$. The operated frequency is 7.6 kHz. The boundary wave excited at channel 1 propagates to terminal 4, indicating that the spin polarization flips from spin down to spin up.

attenuate during propagation due to the loss in air. But the loss would not change the topology of the systems. The influence of air loss on the transmission and topology is discussed in Supplementary Note 7. Figure 4b also gives the comparison of the measured transmissions through a path along the rectangular defect and through a straight one of the same length. The transmission of the two samples agree in the bulk gap, indicating the robustness of the surface modes against backscattering induced by the defect. This result renders the realization of a spin-filtered one-way waveguide.

**Spin flipping in an H-shaped ASCI.** Finally, we show the spin-flipping transport in an H-shaped device to evidently demonstrate the spin nonconservation of the helical boundary states. Actually, the spin cannot flip along a ribbon structure, because

the spin polarization $\langle \sigma_y \rangle$ is independent of the spatial position along a uniform periodic structure without bending. To show the spin-flipping effect, we should consider $\langle \sigma_y \rangle$ in, for example, a H-shaped structure, without uniform periodicity or with bending. In a H-shaped sample, $\langle \sigma_y \rangle$ in the middle ribbon with small width is different from that in the left/right one. Figure 5a shows a schematic of the H-shaped structure of the ASCI, where the width of the left and right ribbons is $W = 20a$, and that of the middle ribbon is $W_m = 1.5a$. The boundary waves with spin-down polarization excited from channel 1 not only can propagate to terminals 2 and 3 kept with the same polarization, but also may transport to terminal 4 with spin flipping, because of spin nonconservation. In Fig. 5b, we calculate the transmissions from channel 1 to terminals 2, 3, and 4 as a function of $L_m$, in which the operated frequency 7.6 kHz is tuned to the passing band of

the middle ribbon. One can see that $S_{41}$ is periodically oscillated, and the boundary waves can flip the spin polarization and reach terminal 4 at the right $L_m$. In Fig. 5c, for a fixed $L_m = 2a$, the measured and simulated field distributions clearly show that the boundary wave excited at channel 1 can propagate to channel 4, demonstrating that the spin polarization flips from spin down to spin up. The H-shaped structure can be designed to act as a spin flipper, and may also serve as a splitter with a switch effect, as discussed in Supplementary Note 8 and 9.

## Discussion

In summary, we have realized an ASCI with a pair of helical boundary states with spin conservation breaking, which is of fundamental interest and opens up an avenue for applications of topological acoustics. This work implies two basic aspects different from earlier works of acoustic topological systems[35–44]: first, the ASCI is induced by spin–orbit coupling, which is independent of any crystalline symmetries, and exhibits gapless edge states on the boundaries. Second, the ASCI breaks the spin conservation, which greatly expands the field of current topological physics, limited to spin conservation. The helical boundary states may have potential applications in innovative acoustic devices, such as topological splitters/switches with high tolerance. It should be noted that a similar acoustic structure has been employed to realize a fragile TI very recently[45], which however is essentially different from the presented ASCI (Supplementary Note 10).

## Methods

**Numerical simulations**. All numerical simulations were performed by the commercial COMSOL Multiphysics solver package. The systems were filled with air with a mass density $1.3\ kg\ m^{-3}$ and sound velocity $343\ m\ s^{-1}$ at room temperature. Because of the huge acoustic impedance mismatch compared with air, the 3D-printing plastic material was considered as hard boundary.

**Experimental measurement**. A subwavelength headphone with a diameter of 6 mm was used to generate acoustic waves. A microphone probe with a diameter of 1.5 mm was used to measure the acoustic pressure field distributions. A network analyzer (Keysight 5061B) was used to send and record the acoustic signals. The dispersions of bulk and boundary states were obtained by Fourier transforming the scanned acoustic pressure field distributions inside and on the boundary of the samples.

## Data availability

The data that support the plots within this paper and other findings of this study are available from the corresponding authors upon reasonable request.

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

## Acknowledgements

This work is supported by the National Key R&D Program of China (Grant No. 2018YFA0305800), the National Natural Science Foundation of China (Grant Nos. 11804101, 11890701, 11704128, 11774275, 11974120, and 11974005), Guangdong Innovative and Entrepreneurial Research Team Program (Grant No. 2016ZT06C594), Guangdong Basic and Applied Basic Research Foundation (Grant No. 2019B151502012), and the Fundamental Research Funds for the Central Universities (Grant Nos. 2018MS93, 2019JQ07, and 2019ZD49). V.P. and S.H. acknowledge support from the European Research Council under the Grant Agreement No. 771503.

## Author contributions

W.D., X.H., F.L., and Z.L. conceived the original idea. W.D., X.H., J.L., V.P., S.H., and Z.L. performed the theoretical part of this work. F.L. and X.H. carried out the experiments. Z.L. supervised the project. All authors contributed to the analyses and the preparation of the paper.

## Competing interests

The authors declare no competing interests.
