## [Peer Review File · Nature Communications]

REVIEWER COMMENTS

Reviewer #1 (Remarks to the Author):

The authors have failed to address my comment. In quantum spin-Hall systems magnetic defects can break the topological protection originating from time-reversal symmetry and create back scattering. I am asking what is the equivalent of magnetic defects in this system. The author just replied on what happens for non-"magnetic" defects. So, I ask the question again.

I am hoping that the authors can clarify this point and include a remark on this. Properly and objectively analyzing the conditions over which the effect can be used is an important aspect especially if the authors really believe that the system will be used in applications.

Response to Reviewer #1:

The authors have failed to address my comment. In quantum spin-Hall systems magnetic defects can break the topological protection originating from time-reversal symmetry and create back scattering. I am asking what is the equivalent of magnetic defects in this system. The author just replied on what happens for non-"magnetic" defects. So, I ask the question again.

I am hoping that the authors can clarify this point and include a remark on this. Properly and objectively analyzing the conditions over which the effect can be used is an important aspect especially if the authors really believe that the system will be used in applications.

Reply: We thank Reviewer #1 for your valuable comments. We are sorry for the misunderstanding previously. Indeed, the influence of defect on the helical boundary states is important in applications.

In electronic systems, the time-reversal symmetry can be broken by the magnetic field, so that a magnetic defect can break the topological protection of the boundary states, giving rise to backscattering. In acoustic system, as the acoustic waves are inert to the magnetic field, the time reversal symmetry cannot be broken similarly by a magnetic defect. But the acoustic waves are non-reciprocal or non-time-reversal in a flowing fluid [e.g., *Science* **343**, 516 (2014)], therefore a 'magnetic defect' for an acoustic system can be created by circulating the fluid component somewhere locally in the structure. Such mechanism has also been used [*Nat. Commun.* **6**, 8260 (2015); *Phys. Rev. Lett.* **114**, 114301 (2015); *New J. Phys.* **17**, 053016 (2015); *Phys. Rev. Lett.* **122**, 014302 (2019)] to realize the acoustic Chern insulator with time reversal symmetry broken. For our system, we can create a 'magnetic defect' on the boundary by circulating the air inside a certain tube close to the boundary connecting the upper and lower layers. Practically, creating circulating flow brings much complexity in design and in experiment. Therefore, compared with the magnetic defect for electrons, which is common, an equivalent 'magnetic defect' for acoustic waves is rare, which

means that the helical edge states observed in our acoustic system are topologically protected, more rigorously than the corresponding electronic counterpart.

We have added the above discussion in the Main Text of the revised manuscript.

REVIEWERS' COMMENTS:

Reviewer #1 (Remarks to the Author):

Thank you for your clarifications. I think they are satisfactory and that the paper can now be accepted.